# Entropy and Cross-Level Orderliness in Light of the Interconnection between the Neural System and Consciousness

**DOI:** 10.3390/e25030418

**Published:** 2023-02-25

**Authors:** Ilya A. Kanaev

**Affiliations:** Department of Philosophy, Sun Yat-sen University, 135 Xingang Xi Rd, Guangzhou 510275, China; ilyak@mail.sysu.edu.cn

**Keywords:** natural evolution, human consciousness, social cohesion, entropy

## Abstract

Despite recent advances, the origin and utility of consciousness remains under debate. Using an evolutionary perspective on the origin of consciousness, this review elaborates on the promising theoretical background suggested in the temporospatial theory of consciousness, which outlines world-brain alignment as a critical predisposition for controlling behavior and adaptation. Such a system can be evolutionarily effective only if it can provide instant cohesion between the subsystems, which is possible only if it performs an intrinsic activity modified in light of the incoming stimulation. One can assume that the world-brain interaction results in a particular interference pattern predetermined by connectome complexity. This is what organisms experience as their exclusive subjective state, allowing the anticipation of regularities in the environment. Thus, an anticipative system can emerge only in a regular environment, which guides natural selection by reinforcing corresponding reactions and decreasing the system entropy. Subsequent evolution requires complicated, layered structures and can be traced from simple organisms to human consciousness and society. This allows us to consider the mode of entropy as a subject of natural evolution rather than an individual entity.

## 1. Introduction

The question of whether organisms experience subjective states evokes extensive theorizing in contemporary science because of predicaments in ethics and technology [1,2,3,4]. A broad spectrum of answers exists, varying from claiming the principal difference from the physical realities to illusiveness [5,6,7,8]. Many recent advances in studying the behavior of organisms have revealed the vital role of experience in their life and evolutionary history, as well as its evident interrelation with processes in the body [9,10,11]. However, in trying to explain why particular changes were preserved by natural selection, it is easy to fall into idealistic speculation and assume a pre-existed design. At the same time, considering evolution as a blind shuffling of possible mutations is a tremendous simplification, outdated long ago [12,13]. Thus, scientific theory should reveal how and why the stochastic changes allowed by the intrinsic regularities of the system provide adaptive advantages and constitute a species [8,14].

Per tradition, the question of the interrelation between matter and consciousness belongs to the sphere of philosophy. This attitude differs from praising it as a foundation of knowledge to rejecting it as logical nonsense [1,15]. With no pretensions to solve this millennial mystery, this study aimed to review some recent advances in theories of consciousness and the evolutionary history of humans [16,17,18,19]. The concept of entropy as a constant tendency to equalize the possibilities of future states is used to synthesize recent advances in anthropology, neuroscience, and other disciplines [20,21,22,23,24]. From the epistemological perspective, one can define entropy as a measure of the uncertainty in light of our ability to predict the future of the system, which provides a link to the “point of view” concept, comprehensively described in [25]. The temporospatial theory, which considers world-brain alignment as a critical predisposition to conscious activity, is used as a framework to discuss how physical systems may have subjective experience [26,27,28,29,30,31,32]. It states that the focal point must be on discerning how the world guides evolution and functioning of the neural system. A recent study argues that such a world-based ontology of self can unravel its prephenomenal features [33]. Therefore, this approach argues for the materialistic understanding of consciousness along with emphasizing the subjective activity. This provides a scientific foundation to rationally explain moral norms, creativity, and free will. In this paper, I suggest that the regular environment can reduce the system’s entropy and guide the system’s evolution from the basic level of orderliness to obtaining the ability for conscious actions and the creative transformation of the living world.

## 2. Methods of Studying Consciousness in Contemporary Science

Studying the neural system can best represent the integration of body functions and guiding behavior because of its plasticity and the possibility to have reports about the subjective states in humans. However, there is an ambiguous situation in contemporary consciousness studies that still lacks a comprehensive theoretical outline and research strategy [1]. It is possible to highlight an opposition between understanding neural processes as computations between separated physical entities (neurons or their networks) and more general computations occurring within neural spaces [34]. Meanwhile, explanations of subjective experience are usually represented by a set of theories that use different conceptual frameworks [31,35,36,37,38,39,40]. However, many of these studies have paid little attention to understanding the neural structure in the context of naturalistic behavior and the evolution of species [11]. Such simplification is predetermined by examining the neural system using an analog with a computer, whose function can be analyzed as stimulus-evoked activity in isolation from its evolutionary and life histories [32]. In turn, this view is possible because all modern computers are artifacts, in which external sources, rather than self-organization, predetermine formation, sustainability, and linear development [2,3]. This impels understanding neural systems in a mechanistic paradigm, where behavior results from the computations performed by the subsystems. However, the prior task of an organism’s neural system is to control behavior and increase chances for survival; thus, it cannot be excluded from examining how these behaviors are experienced [10,39]. Simultaneously, natural behavior and evolution of species cannot be excluded from the process of aligning with the rhythms of the environment and the linear development of systems [41,42]. According to this task, the material represented in this research is chosen to constitute a comprehensive outlook on the connections between subjective reality, consciousness, and evolution of the species rather than to present a review of existing theories of consciousness. 

### 2.1. Global Connectivity, Integration of a System, and Cognitive Abilities

Among the explanations of consciousness, the global workspace theory is possibly the most integrated with studies in other disciplines and grounds on a vast empirical foundation [43,44,45,46]. Recent advances in this theory have focused on the evolutionary explanation of neural system functioning. In particular, a comprehensive study on the hominization of the brain highlights the crucial role of a complex connectomic structure [47]. It has been suggested that boosting the size of the human brain causes an increase in the quantity and complexity of connections between neuronal areas. Increased connectivity could deepen the difference between the states of the whole system and straighten the organism’s ability to discriminate them as comprehensive states of subjective reality, which increases the informational richness of the system, see [24]. The enhancement of cognitive abilities was an effect that favored chances for survival and reproduction, which were simultaneously preserved by natural selection [17]. In summary, these allow the consideration of cognitive abilities as an emerging systematic feature of the whole neural system rather than a byproduct of merging sub-system computations [48]. This statement is not as trivial as seen because many studies of cognition in neuroscience still try to construct an ideal experimental state to isolate the functioning of particular areas, such as those responsible for visual perception. Considering them embedded in the organism’s behavior can resolve many difficulties [49,50].

The ability to discriminate between and within experienced states is critical for control over behavior, an important part of one’s cognitive skills closely related to working memory [51,52,53,54]. In addition to arguing about the nature and definition of this concept, it has been demonstrated that various neural areas are involved in its functioning [55,56,57,58]. Furthermore, recent studies have demonstrated the interconnection between one’s ability to operate information in an instant action and the general state of one’s body, such as tiredness [59,60,61]. In natural activity, it seems that the capacity of human working memory is limited within operating of three to four items simultaneously, which allows activity with tools and complex future planning [52]. This is supposed to be sufficient to invoke the production of symmetrical tools and the emergence of creative culture [53,62,63]. The results of computer modeling suggest that the limitations of working memory can be caused by interference between the signals [64]. However, the quantity of modeled signals was relatively small and incomparable with the number of neurons in an organism’s neural system. Thus, it must represent the functional division of neural areas in the brain, which is consistent with the supposition of the global workspace theory that the sophistication of connectomic patterns is among the most meaningful events in human phylogenesis [47].

The global workspace theory results and working memory studies agree with the integrated information theory. The latter claims that the intrinsic qualities of the conscious state are specificness, integration, and exclusiveness [65,66,67,68]. This theory suggests that the decisive trait of the system that allows these qualities to emerge is that its overall information must exceed the sum of the information of its parts. A system composed of causal feedforward mechanisms cannot produce additional information regardless of its structure. Thus, the system’s design must include at least some recursive elements in which the previous state of the entire system predetermines the reaction to the external stimuli. These results were obtained based on mathematical and computer modeling and met some experimental verifications. However, the extraordinary complexity of the suggested mathematical apparatus of multi-spatial geometry makes it impossible to apply it to many items as neurons in the brain of almost any known organism. It does not seem necessary that these recursive mechanisms must be realized at the simplest elements, such as neurons. That said, it can also be represented at the functional level of the neural system’s organization. For example, measuring information on the base of several hundred nods may met predictions of the integrated information theory about the timescale of conscious human perception [69]. This interpretation seems to be coherent with the model of flexible working memory and the hypothesis of brain hominization [47,64].

### 2.2. Time, Rhythms, and Sustainability of a System

The recursive organization is a decisive requirement for the emergence of additional information in the system. Its ability to experience subjective states involves the critical role of time and particular requirements in the system’s design [31]. The speed of a set of feedforward mechanisms is restrained only by the limitations of the physical mechanism, so that the aim can be realized as quickly as possible. This is how most contemporary models of naturalistic behavior work. Nevertheless, it implies a tremendous overcomplication of the system, making it extremely energy-ineffective [65]. A large number of basic units of the neural system would have made it impossible for any ordered activity based on the one-level architecture of computational units. Thus, even modern machines and programs already require a multilevel design where control over the system’s functioning operates with statistical data gathered from the basic layers [70]. However, the probability of such a complicated system’s emergence by natural selection is ultimately shallow if it ever exists. In the case of the occasional formation of such a mechanism, they would have been extremely vulnerable to changes and mutations, which inevitably occur. Each complication of a system with a higher control level advocates for the role of intentional and artificial design. Nevertheless, the distinctive feature of the organism is that it emerges as a result of the evolutionary selection of systems that align with the world best, rather than implementing a pre-existed plan, e.g., any artifacts. Thus, any organism’s neural system and subjective reality result from natural selection for the fittest behavior under given circumstances. This implies that including the system’s previous state in its processing must be realized in the most resource-effective manner [71,72,73].

The neural system of animals with complex behavior contains many sub-domains, which are the product of a long evolutionary path and unceasing natural selection of the ones that best fit the environment [74,75]. It has been suggested that the layered brain architectures can “scaffold themselves across multiple timescales, including the ability of cortical processes to constrain the evolution of sub-cortical processes, and of the latter to constrain the space in which cortical systems self-organize and refine themselves” [76]. Furthermore, it was demonstrated that one’s ability to operate on incoming signals is highly dependent on various global rhythms. The frequencies of the neural system’s spontaneous activity constitute a receptive window that predetermines which stimuli can be perceived and which will be omitted by the organism [74,77,78]. More global cycles are caused by overall tiredness and finally constitute a daily schedule by means of the need to rest. All of these can be comprehended in the concept of “temporal nestedness” that was suggested in the temporospatial theory of consciousness on the foundation that intrinsic or spontaneous brain activity is a necessary predisposition for any stimulated processing in the neural system [26,29,30,31,32]. One of its main principles is that slower processes are stronger than faster ones because only the former can guide the latter [33].

Other fields and research confirm this proposition. First, a breakthrough study in brain-to-brain synchrony has already sketched a method for decoding the content of consciousness on the basis of characteristic patterns of the default mode network functioning [50,79]. The success of this approach to brain reading is due to its attention to relatively stable and general patterns of neural activity rather than attempts to find a distinct computation that differs from subject to subject. Second, advances in transcranial magnetic stimulation have demonstrated an effect on participants’ cognitive abilities and could even temporarily restore the capacities of working memory in the elderly and other impaired cognitive skills [80,81,82]. This demonstrates how subjective experiences can be affected by modifying the intrinsic rhythms of the neural system. Third, this might not be that evident; however, few perspective studies in cross-cultural neuroscience demonstrate how the particularities in brain structure predetermine an individual’s cognition and the mode of their interaction with the natural and social environment [83,84,85]. They reveal the focus of Western culture on analysis and self-sufficiency while favoring synthesis and social cohesion in Oriental cultures [86,87,88]. This difference may be grounded in a relatively higher rate of default mode network activity in the former and the significance of the amygdala, basal ganglia, and temporal lobe for emotional control in the latter [89]. Since humans can form an environment according to their needs, these cultural traits demonstrate how qualities obtained during phylogenesis can not only be an advantage or disadvantage during natural selection but also become a source of creative approaches to reality that affect the future development of the species [85,90,91].

### 2.3. The Temporospatial Framework for Studying Neural Systems

Introducing the concept of “temporal nest” or “scaffolds” as a predisposition to align with the world and conscious behavior assumes that areas responsible for the intrinsic activity in the brain outperform and guide those responsible for processing stimulations [32]. Since the basic units of the neural system do not demonstrate significant qualitative differences in their power, while equal faster frequencies imply higher energy, there must be qualitative dominance of the former over the latter. Empirical data verified this theoretical requirement. Although establishing particular areas responsible for intrinsic activity is a matter of ongoing research and discussion, it can be claimed that most of the neural system is involved in its production [72,92,93]. This requires a complicated connectomic pattern that integrates neural subsystems under one principle of orderliness [47]. From the perspective of the entire system, these subsystems can be considered processing units rather than single neurons. Considering the neural system as the “mixture of experts” [94] explains how their collaboration and competition define a particular form of experience and preferred behavioral strategies [95]. Such a design allows for flexible adaptation to changes in environmental regularities and facilitates plasticity of the behavior [96,97,98]. Human evolutionary development warrants mastering the ability to control intentions, which implies the critical role of social environment [99]. However, the computational abilities of the neural system and the optimal response time to an event predetermine the limits of complicating such a system. This is coherent with the model of flexible working memory, which assumes that the richness of the subjective state is constrained by the interference of neural areas [64]. At the same time, aligning with the world engages the whole body. It predetermines those simple strategies are effective under certain conditions and when the reality is too complicated to anticipate all its possibilities [100]. Thus, evolutionary history defines a species’ cognitive abilities’ optimum and organization in a multi-level system [101]. 

Figure 1 provides the author’s elaboration on the recent advances in considered theories, and suggests theoretical demarcation between concepts of subjective reality and consciousness (a). Many studies use measuring the entropy of subjective reality in a resting state to predict the presence of conscious awareness [24,102,103,104]. Being synthesized with conceptualization of the inverted relationship between neural mechanisms and functionality of mental features [27], this allows to suggest an inverse relation between entropy of neural system and subjective reality (b).

Entropy has at least two foundations. First, the stochasticity of any system comes from the fundamental principles of the existing reality and thus cannot be eliminated. Second, entropy, as an intrinsic tendency to disseminate any orderliness, is vital for an organism’s existence because it helps avoid lodging in one state and returns the homeostatic state from which any required action is equally possible. An interesting discussion about the role of serotonin as a necessary condition for adapting to new circumstances that must be favored by natural selection is given in [24]. Thus, the sustainable functioning of any relatively complex organism depends on preserving the balance between aligning with the world that prescribes future actions and the system’s uncertainty that allows adaptation to the changes [22]. Integration of system elements and coherence between intrinsic activity and stimulations reflect a decrease in entropy. In contrast, the system’s disintegration and contradiction with perceptions causes an increase in entropy [23]. Both these processes are necessary for the species’ survival because the former provides adaptation, while the latter guarantees alertness and plasticity of behavior. This is why even the state of ultimate coherence and satisfaction cannot last long, or it will just cause the death of individuals and the extinction of the species.

Maintaining the required proportion between the stability and uncertainty of the system, as well as the integration of its elements, relies on a particular physical mechanism. That is, in studying the neural system, the most critical requirement is that the entropy of the integrated system must be counted on the basis of the elements’ traits, as a product of subsystems, rather than the sum of their entropies that can be neglected. Correspondingly, after integration into a higher order system, its entropy must be counted on the basis of the sub-system’s traits and not as the sum of the sub-system’s entropies. This principle is used in the model of flexible working memory, which explains the limitations of cognition by the interference between relatively small amounts of agents [64]. Otherwise, there is no way that a system with so many elements as the human brain can align with the regularities of the environment because of the high noise that comes from the mass of the sub-system units [69]. Therefore, the feedforward design can hardly introduce additional information because its entropy is merely the sum of its element entropies [65]. Thus, the higher layer of orderliness should use physical principles of integration other than the lower one. This requires that subsystems are represented by the product of their element activity, rather than the same principle of interaction between elements; see [71].

The discussed data can be used to argue that the interference between intrinsic and stimulating activities of the neural system is a physical correlation of the experienced subjective state [99]. This assumption is supported by empirical and theoretical evidence that interference between these neural subsystems may predetermine cognitive abilities and the complexity of the subjective state. Although direct connectomic interactions play a vital role in neural system functioning [47], it is possible that, in some cases, electromagnetic interference can serve as a medium for binding activity in the separated areas. The most interesting example is the split-brain case, in which the unity of subjective reality and working memory capacity is preserved despite the lack of direct connections between brain hemispheres [105]. Furthermore, the interference pattern of the entire neural activity can also affect single neurons, providing the required orderliness and decreasing entropy of this basic level. It can be considered an analog of hormonal regulation that affects the current state of the whole body and predetermines many of the behavioral responses to the world. Thus, in an organism’s set of vital organs, the neural system is a sub-system that controls the ongoing behavior and response to the current environmental effect, based on past experience [9,48,97,106]. It operates with possibly faster rhythms of instant action that have control over many other processes. Although this control cannot last long, it is usually sufficient to guide instant actions, including prosocial behavior, by assigning the corresponding valence to the perceived objects [95,107]. Therefore, it can be theorized that the behavior of the most known species is guided by the top level with several competing subsystems and demonstrates a correspondent entropy that provides the required level of alignment with the environment in a scale-free way, see [33].

## 3. Results of Considering Subjective Reality within the Evolutionary Framework

Evolutionary origin of species may be one of a few undisputed postulates of the modern scientific worldview [44,108,109,110]. Although the particularities of the evolutionary theory and the sequence of species development are subject to reconsideration, the idea of generative development of complicated structures from simpler ones is generally accepted in all disciplines. This assumes that all complex entities are the result of some processes at a more fundamental level and must have some simpler predecessors in their past. However, the more complicated a system, the more vulnerable it should be to stochastic changes and an increase in entropy as a tendency to dissipate any structure [111]. Nevertheless, the development of organisms from simpler to more complicated ones demonstrates that the evolution of orderliness is a natural tendency that must be grounded in the basic principles of reality.

### 3.1. Sustainability of Regular Changes and Linear Evolution of a System

In addition to the ongoing discussion on the mechanisms that cause the origin of life as a homeostatic system, there is a simple observation. If ongoing events do not demonstrate any regularity or rhythm, it is not possible for any system to adapt to them because adaptation assumes anticipation and correspondent change of the own processes, which becomes the foundation for the behavior [112]. Irregular events can provide only favorable or unfavorable conditions for some assemblage of things but cannot invoke the emergence of any structure or system. Thus, only environments with strong regularities may cause the emergence of an adaptive system whose cycles reflect ongoing changes [113,114]. The fundamental gravitational, electrical, and other interactions may form an aggregation bonded by inner reactions, some of which receive reinforcement from the environment [108]. Within this system, the probability of supported reactions increases, whereas the possibility of others decreases. The outer world introduces constraints on the excessive freedom of this system, pushing it towards “criticality” [20,115]. This process opposes the natural tendency to dissipate any structure that originates from the stochastic movement of things. This can also be described as a decrease in system entropy.

While the system’s lifetime depends on maintaining entropy under a particular threshold, an unceasing process of such simple system arousal and decay must lead to the appearance of relatively sustainable systems with a cycle of homeostasis [116]. After obtaining this ability, such a system can be considered an “organism” that integrates many assemblages under a particular orderliness [23]. The simpler the regular changes in the environment, the greater the potential to invoke the corresponding anticipation. The second term indicates that the frequency of these changes must be coherent with the foundational reactions of this system. That is, an organism’s ability to align with the environment is the adaptation process that is effective only when the entropy of its complex behavior is coherent with the regularities of the environment [112]. The outer world plays a vital role in the organism’s functioning and guides its future development. For example, an iteration of the two environmental conditions has the greatest chance of invoking corresponding responses in organisms, which was observed long ago; see [117]. On Earth, this change in night and day affects almost any ecosystem. The orderliness adopted by the system reflects environmental changes and can be qualified by the corresponding entropy mode and value, see [24].

Effectively adapting a system to the simplest regularities prolongs its temporal longevity and spatial growth [44,113,118]. However, physical principles introduce limitations on the quantity of components in a particular system design, and irreducible entropy leads to the dissipation of any structure. These constraints imply that a system with sufficient sustainability either replaces some of its components or may be divided into separate systems with the same orderliness. The latter is a foundation for reproduction so that the mode of anticipating reality may be preserved and subject to natural selection. The latter comes from the stochastic changes in each generation of this system so that the fittest ones reproduce better and subsequently replace their predecessors. Therefore, the intrinsic entropy of the system not only dooms any system to annihilation but also serves as a vital predisposition for the evolution of its mode of orderliness. Figure 2 demonstrates conceptualization of evolutionary development.

Natural selection for better alignment with the world assumes constituting particular cycles of this system’s comprehensive states, which reflect regularities of the environment and prescribe behavioral responses to anticipate them. These states result from a long evolutionary development and are thus modified to better represent possible events and corresponding reactions [39,75]. In organisms with complex behavior, this function is performed by the intrinsic activity of the neural system that provides alertness to the stimulation by needing meaningful income data [78,97]. Significantly, most natural behavior requires the involvement of the whole organism, as it happens in the fight-or-flight response [96,119,120,121]. This can predetermine the quality of the exclusiveness of the subjective reality’s state, which is usually overwhelming and changes only over time. This is consistent with the suggestion of the integrated information theory that a system with a surplus of information must experience subjective states of being, which integrate and represent all processes occurring in the body [65]. Such a design facilitates the reaction to the environmental effect rather than if feedforward mechanisms are used. The surplus speed is accompanied by increasing efficiency because simplifying the reaction saves energy and broadens the range of possible states [71]. Natural selection should support this critical advantage, which becomes a driver for further sophistication of the organism’s control system and experience [39]. This opens the vista for the subsequent development of system complexity and unceasing evolution. As the system evolves, it can adapt to more complex rhythms to comprehend them in cognition and apply mastered principles to the broader scope of ongoing events [97,122,123].

### 3.2. Increasing Complexity of the System from the Organism and Up to the Society

If the environment is ultimately regular, the most effective structure integrates all activities in one top level that reflects the outer world and predetermines the behavioral response. This may be the level of the simplest unicellular organism, which interacts with reality in quite a limited way so that the basal cognition they demonstrate is sufficient [118,124]. However, sustainable development of the system involves prolonging its temporal and spatial longevity, which not only increases the entropy of the system but also causes facing regularities in the environment, which have had no sense before [125]. Hence, anticipating new rhythms implies a new round of adaptation and rearrangement of the system’s design [75]. Such an adaptation can introduce a higher level of orderliness or several competing control subsystems within a single level (see Figure 2 above). A more stable environment complicates the organism’s structure and decreases the system’s entropy. Complicating the body structure constitutes a new level of orderliness encompassing new cycles of environmental change, usually slower than those from the basic layer, such as yearly changes overwhelming daily changes. This helps reduce the entropy of the system and enhance its sustainability. In turn, a more variable environment should favor the sophistication of the existed top layer responsible for the behavior. Introducing competing subsystems within one level of orderliness causes competition that disturbs simple orderliness and inevitably increases the system’s entropy, see [94].

Anticipating regular changes in the environment constitutes the basal form of cognition of an organism that is critical for sustainable existence and involves the emergence of vast behavioral strategies. Among these, social behaviors are of the utmost interest because they allow us to observe the process of the system’s formation in dynamics [101]. However, it is necessary to provide a definition of social systems and to demarcate them from others. An assemblage of things can be very sustainable and demonstrates inner connectivity between parts. However, this situation exists only because occasional circumstances, such as the oversaturation of the required elements, favor the establishment of these interactions. Facing environmental changes, these things do not perform any behavior to preserve their bonds, which differs from ordinary activities. Thus, such assemblages can be very sustainable but not social. The opposite situation occurs in a unified system of organisms. A canonical example of reciprocal behavior is soil bacteria, which face a lack of nutrition, form a fruiting body, and even demonstrate kin discrimination [126]. Social insects, such as ants or bees, exhibit much more complex behavior that is completely guided by the group’s needs. The specialization of social units for reproduction and survival allows us to consider them as a specific form of an organism in which parts can be separated in space rather than a society in its own sense [127,128]. The comparison between assemblages of things and unified organisms implies that social relations emerge between organisms whose behavior is intrinsically connected with the whole system but is not strictly predetermined either by the environment or by inner processes in their body. This increases the stochasticity of the system but simultaneously opens a vista for the formation of a system more complex than just the sum of the processes in its elements and environment [42].

It would be a mistake to consider social systems as something very different from their evolutionary predecessors. Thus, humans demonstrate the most complicated social interactions, in which analogs and prerequisites can be found in many species [129], but first of all in primates [18]. It has been reported that sharing emotional states such as anxiety is an effective strategy of communication in a group because it reduces the time of detecting the danger and thus increases the chances for survival of any individual [114]. Furthermore, exploring the environment is necessary for control over resources, but it is extremely time-consuming with no guarantee of success. The transmission of knowledge about the environment between collective members allows for more effective foraging and rest [107]. The higher the cognitive abilities of an individual organism, the more sophisticated the strategies for acquiring knowledge from the other. Notably, in many cases, bonds between individuals are more important than their kinship, with bonds between sexual partners being among the most stable and efficient [16].

### 3.3. Human Society and Transformation of the Reality

Social structure presumably plays the most critical role in humans because of the extraordinarily long period of child immaturity and the need for alloparent rearing of offspring [130,131,132]. Maturing juveniles corresponds to a better time for learning new skills [133,134]. This facilitates the transmission of knowledge between generations, which is critical for preserving the achievements of culture and its future development [135,136,137]. The formation of cognitive abilities and neural system structure corresponds to the task of subsequent incorporation into a particular society. The high diversity of social requirements involves the necessity of mastering them during life history rather than being an innate instinct, as found in social insects [138,139]. This predetermines the formation of human consciousness as a particular skill of controlling behavior at the level of intention [99,140,141,142]. Altogether, a balance between predetermined body structures and socialization can be found in humans [98]. The former follows the rhythm of ontogenesis, while the latter is responsible for adapting to the local environment and the linear evolution of the system. Human life history is a highly divergent realization of the sustainable system’s sequence, which is possible only because of the unique traits of the human neural system that grant a lot of freedom to an individual’s subjective reality. Figure 3 demonstrates a coordinate system that combines physical, biological, and subjective perspectives on the system’s processes in the evolutionary framework.

These results suggest that the structure of an organism’s orderliness results from the subsequent and relatively smooth evolution. However, the integration of subsystems under the new and higher level of orderliness is usually linked to obtaining new abilities imposed by the extraordinary change in the environment. The most critical challenge for a species’ survival is that the spurt in their development is accidentally accompanied by significant environmental changes [132,143]. This could have happened with humans when the subsequent evolution faced a climate switch that almost led to the extinction of the species [144,145]. The changeable environment introduced a higher requirement for cognitive abilities, which resulted in formidable complications of the human neural system and hominization of the brain. Increasing the connectome complexity caused an outstanding diversification of the current state’s content and enhancement of the working memory by up to three to four items. This evolutionary leap favors tool operations and communication abilities [63,146]. At the same time, it prolongs immaturity, invoking the need for alloparent care because other processes within the body remained the same. Together with environmental challenges and social selection, maintaining social cohesion in massive collectives became critical for the survival [17,147]. Because of the high variability of individual primate behavior, this was possible only by mastering self-control over their intentions in light of the particular social traditions unavailable for other animals [43]. One can consider this consciousness narrowly, which affords knowledge accumulation and the culture’s emergence [99].

The control over intentions assumes that the body’s subsystems are united under this higher level of orderliness. Although such control cannot last long if there are basic needs such as hunger or danger, it is usually sufficient to provide the required behavior when facing the needs of the collective [140]. Thus, mastering the ability for conscious activity affords the consideration of individuals as entities with particular traits and neglects the entropy of their body, just as it happens in the neural system that provides the unified state of subjective reality based on interference from many neural areas. This can be an example of organism’s scale-free alignment with the world [33]. As discussed above, only slower and more powerful rhythms may become scaffolds for faster frequencies and constrain them [32,76]. Introducing the culture and historicity that outperform any individual’s lifetimeallows control of the body’s intentions and reaching the required level of social cohesion [148,149,150]. Among the known species, only humans can master such abilities to date.

Furthermore, conscious activity allows long-term planning and the creative transformation of the environment. Such a modification of entropy already exceeds the limits of the organism’s body or collective [71]. This creative attitude to the environment from human consciousness represents another coil of natural evolution as emerging and dissipation of systems invoked by the world [44]. Therefore, one can consider the system’s entropy as a real agent of natural selection rather than genes or other things from the basic level of orderliness.

## 4. Discussion of the Research Limitations and New Vistas

Summarizing the discussed evidence indicates that the neural system is a sub-system of a complicated organism that plays a critical role in aligning with the world. In turn, the neural system also demonstrates a multilevel structure. Its basic units’ gait into the neural areas responsible for the particular processing and up to integration into the unified state of subjective reality that guides behavioral response to the environmental effect. Such a system can be evolutionarily effective only if it can provide an instant alignment between subsystems in light of the incoming effect. The temporospatial theory argues that such alignment is possible only if the system performs its intrinsic activity, which is adjusted for compliance with the regularities of the environment and modified by stimulation. I hope that the discussion will invoke debates on considering human consciousness within the evolutionary framework and how the concept of entropy may be used to integrate physical and subjective realities. The limitations of this study come from the limited abilities of the author and the set objectives.

This research was focused on philosophical issues and aimed to present them as a review rather than introducing the same ideas via experiments, modeling, or calculations. Due to the vagueness of the theme and the existence of many theories of consciousness and methods to count entropy, any attempt to consider raised issues more particularly would have been a voluntary choice of some mathematical or research model caused by the author’s preferences and knowledge, with a pretention for some objective inquiry. An effective method to deal with the entropy of the system’s physical and subjective states [22,24] is a challenge that must be resolved by the work of the scientific community rather than by individuals. I suppose that the method and formula must be as simple as possible and connected with the known principles of physical reality. Furthermore, it may be reasonable to consider the entropy of a system in its interconnection with the regularities of the world rather than a closed entity. More profound elaboration of this issue should be based on the philosophical background presented in [33].Considering entropies of the neural system and subjective reality allows us to find new methods of estimating the content of consciousness from the third-person perspective (see Figure 1 above). However, this requires profound empirical research using neuroimaging and other methods to obtain reliable data. To minimize possible bias in the evaluation of subjective reality’s functioning, it is critical to pay great attention to cross-cultural and other diversities, which are relatively poorly studied yet [151,152,153]. As discussed above, the top level of the human orderliness is consciousness. It must be mastered during an individual‘s life by including society and adopting its regularities, which are slower than simple behavioral functioning [84,88]. Thus, particular differences in the neural system across the human population may be best seen in the context of temporal functioning rather than the spatial organization of the brain connectome, see [31]. Hence, the temporospatial theory of consciousness [26,27,28,29,30,31,32,33] seems to be the most appropriate framework for future research on human consciousness, freed from the unintentional bias grounded in considering a part of the population as an etalon.Despite attempts to highlight the critical role of time, its understanding is a great challenge that comes from one of the most foundational qualities of subjective reality: its exclusiveness [66]. Complicating the current state and quantification of time is a natural tendency for the extensive development of the current level of orderliness. At the same time, any attempt to comprehend regularities of the higher order requires dealing with slower frequencies, which are difficult to reach by means of the current tools of perception and action. The development of any system, from organism to culture and knowledge, follows the same process as natural selection with the available regularities of the environment [132]. In turn, the latter requires that the existing system be sufficiently rich in meeting them and warrants sustainable development for future development. Thus, the concept of temporal nestedness suggested in temporospatial theory may have a much wider application that describes the functioning of the neural system [26,27,28,29,30,31,32,41,42].Last but not least, considering the creative abilities of humans to modify their living world raises ethical and philosophical questions [3,4]. Any complex system is a product of aligning with the world; however, it is also affected by its functioning. From a physical perspective, preserving a system’s orderliness facilitates its environment’s entropy. However, the basic needs of the organism are the product of the natural selection of the species and, thus, tend to preserve the balance with reality. As the product of social selection, human consciousness reflects more rapid changes and can introduce meanings above natural needs, which tend to modify the environment much more dramatically [90]. Thus, the ability for intentional activity in a cohesive collective allows humans to control entropy not only within the group but also outside it. Recent social disturbances have highlighted these dangers [154,155]. The outcome of our actions depends on the adopted intentions and goals, which may be both within the system and the world. Significantly, the borders between them are determined by one’s consciousness: it can be an individual body, family, society, or the whole world perceived as a unity. Determining which goal will be chosen is a matter of current choice.

## 5. Conclusions

In a regular environment, any occasional assemblage of things is subject to natural selection for coherence with the world by obtaining surrounding orderliness. Thus, the latter’s effect allows the maintenance of the entropy of the former below a particular threshold and causes the emergence of a system that can preserve homeostasis, anticipate upcoming events, and reproduce under suitable circumstances. This demonstrates the scale-free nature of the world’s orderliness.Interaction with the world and natural behavior involves the entire system, requiring all its components to be united in a cohesive motion that constitutes a specific subjective state. Therefore, each anticipative system can be considered to have a particular subjective reality with its cycles of activity and level of entropy that is coherent with its living world.The evolution of systems involves facing new regularities and starting a new round of adaptation, either introducing competitive subsystems of the same level or mounting a higher level of orderliness. The former complicates the behavior and increases the entropy of the system. The latter is linked to obtaining new abilities, which usually invoke changes in the organism’s structure. This decreases the entropy of the system and can even lead to the emergence of society.The rhythms of the higher level must outperform and guide the activities of the subsystems. This assumes that the higher level constantly performs its intrinsic activity, in which frequencies are slower but involve more units than any sub-system. The human neural system demonstrates all of these qualities and grants exclusive access to the contents of subjective reality. The extraordinary evolutionary history of the human neural system allows one to master the ability of conscious action. This is an achievement that must be gained during the life history of an individual, and the duty of society is to provide opportunities for this.The framework of the temporospatial theory allows us to understand human consciousness in the context of natural world evolution. This demonstrates that alignment with the living world is necessary for achieving coherence between the environment’s regularity and the system’s entropy. However, the emergence of human consciousness may be a new coil that allows for intentional change in entropy within the system as well as in the environment. This increases the responsibility of the species to master conscious action during their lives.

## Figures and Tables

**Figure 1 entropy-25-00418-f001:**
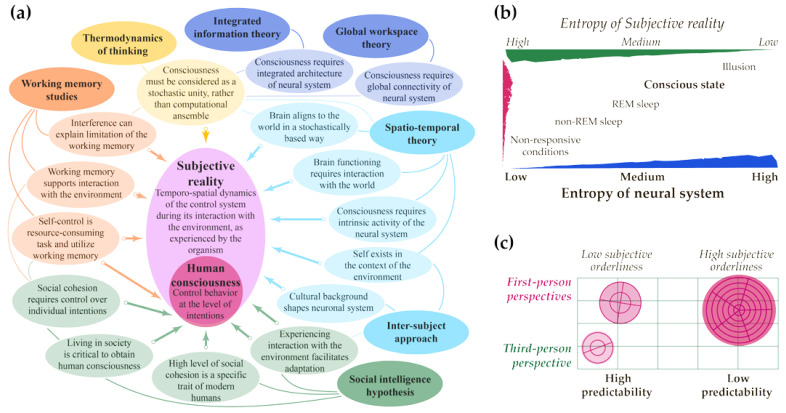
(**a**) Theories and advances in studying consciousness synthesized in the concepts of the subjective reality and human consciousness. (**b**) Inverse relation between entropy of neural system and subjective reality. Entropy is a degree of the disorder of a system and complexity of its dynamic changes that can represent predictability of neuronal system (as objectively measured) and subjective reality (as experienced by an organism). The higher is unpredictability of the neuronal system, the more consistent is subjective reality, and vice versa. A particular level of consciousness reflects the alignment of the brain to the world and can be characterized with a corresponding level of entropy, measured both from the third- and first-person point of view. Source: author’s Figures 2 and 4 from reference [99]. (**c**) Interrelation between points of view and subjective orderliness of the agents. Third-person perspective is constituted as approximation of the involved system’s activities, which assumes that the ones with lower orderliness are easier to predict, and vice versa. For the detailed analysis of the point of view concept, see [25,33].

**Figure 2 entropy-25-00418-f002:**
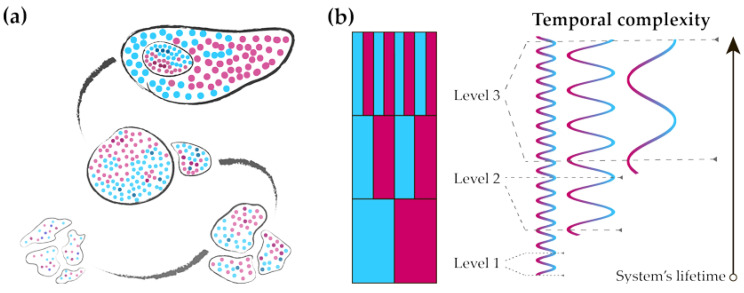
Alignment of an organism (**a**) with regularities in the environment (**b**). An increase in the system’s complexity allows it to deal with increased challenges, which are constructed over the existing background. The colored dots represent temporal dynamics of the organism’s global states rather than its spatial structure. The simplest iteration of the two opposites can produce eight possible global states at the third level.

**Figure 3 entropy-25-00418-f003:**
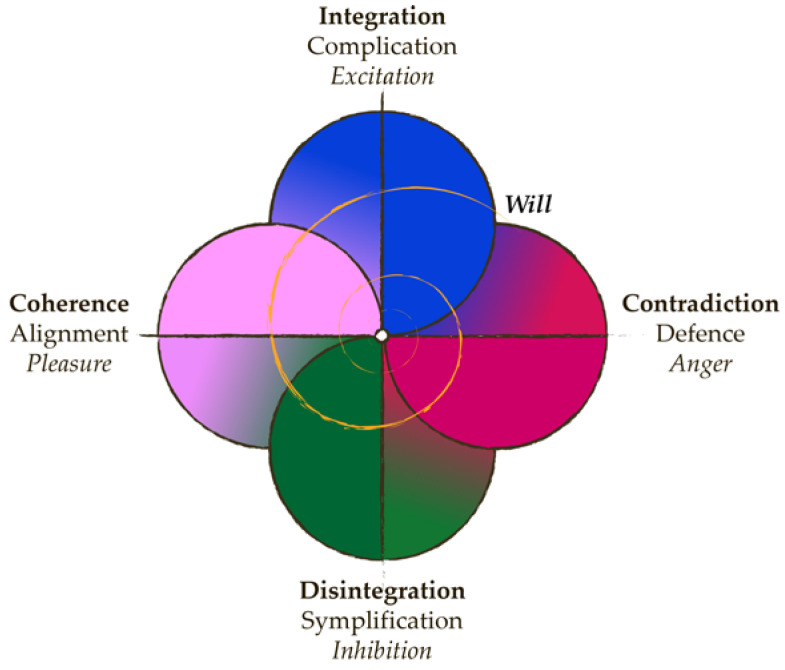
Coordinate system combining physical (bold text), biological (regular text), and subjective (italic text) perspectives on the systems’ processes. Entropy increases from the center to periphery in all four directions. Integration of a system and increasing complexity of the body is critical for evolution. Disintegration is the basic process for (re)production and resource consumption. These two aspects form the foundation of homeostasis and represent increases and decreases of the systems’ energy. This predetermines their experiences as excitation and inhibition. Coherence and contradiction are two possible types of interaction with the environment. Reaching coherence with the income stimulation causes alignment between the world and the system. This reinforces the inner processes in the latter and invokes feelings of pleasure. Facing contradictory event creates danger, which causes a self-defense reaction and feelings of anger as a signal of the body’s readiness to choose between fight or flight. Coherence and contradiction are required for adaptation to the environment. An individual with the corresponding body structure and required training can use their will to intentionally change their state. This must follow principles of change, which usually uses the opposites to overcome the present state and invokes spiral movement. Furthermore, this is difficult. Source: author’s Figure 3 from reference [99] with changes.

## Data Availability

Not applicable.

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
