# Peer review of "Entropy and Cross-Level Orderliness in Light of the Interconnection between the Neural System and Consciousness"

_entropy, 2023, doi:10.3390/e25030418_

Round 1

Reviewer 1 Report

The reviewed paper is an interesting voice in the discussion on explaining the issue of consciousness from the perspective of neuroscience and the theory of evolution. The author suggests that consciousness evolved in response to the need for biological systems to minimize entropy. I believe that the author's starting point is correct, but I have serious doubts about the implementation of this task. My main complaint is that the author jumps to the next conclusions too quickly and in an unjustified way.

For example, it is not clear what subjective states, experiences or reality are, which the author associates with consciousness. Whether the existence of such states implies consciousness or not. For example, the author states that "subjective reality result from natural selection for the fittest behavior under given circumstances" (3). This statement seems to be true in a sense, but it is too general when formulated as such. What does it mean that something results from natural selection? In what sense? In this mode of speaking, basically anything can be said to be adaptive. However, it is so general that in principle it can mean many things, and probably in one sense subjective states are the result of adaptation and in another, for example, how the retina processes visual information.

My other doubt concerns the dependence of entropy, homeostatic processes and higher biological or adaptive functions. There is a huge literature (which the author skillfully refers to) which in a more or less controversial way (cf. Friston et al.) proves that homeostatic mechanisms can be understood in terms of long-term minimization of entropy, resp. free energy (both thermodynamic and information sense). However, it seems doubtful to treat many biological mechanisms or functions as homeostatic. What I mean is that the explanation of high-level subjective states, as defined by the author, cannot be explained analogously to low-level homeostatic mechanisms. At one time, Peter Godfrey-Smith (1996) pointed out that there are significant limitations associated with talking about homeostatic mechanisms. Namely: the actual homeostatic mechanisms consist in maintaining the stability of those organic properties that are not trivially related to survival. That is, when some complex activity of an organism contributes to survival, it is truly homeostatic only if there is some intermediate organic property, that is, that this complex activity contributes to the maintenance of homeostasis in that intermediate property, insofar as it contributes to real contribution to survival. For example, the intelligent use of fire by humans to maintain body heat is therefore not a true case of homeostasis implemented by some cognitive mechanism. It is therefore a complex organic ability that enables a human to maintain a constant body temperature in changing environmental conditions, thus enabling survival in these conditions. But there is no reason to think that everything that happens through cognition is trivially homeostatic. This means that the mere explanation of homeostatic mechanisms is not enough to explain what, for example, are the high-level mechanisms responsible for the emergence of consciousness.

I also miss a greater involvement in the classical philosophical literature on biological functions (e.g. Millikan, 1984; Bickhard, 2003 and so on) as well as the latest on the role of constraints (e.g. Bechtel, 2019; see also Pattee, 1991) or biological autonomy or dissipative structures (e.g. Moreno & Mossio, 2014; Ruiz-Mirazo & Moreno, 2004). These analyzes, I believe, would contribute a lot to the discussed issues and would help the author in consolidating many of his (too hastily) adopted theses.

It is too hasty for me to move from analyzes of the stochasticity of biological (or basically physical) systems to the constitution of social systems (8-9).

Sentences like this: "An individual with the corresponding body structure and required training can use their will to intentionally change their state" (10) are unjustified and completely unclear, i.e. the content of the paper does not justify them and they do not follow from it.

The author describes his paper as a review. I have big doubts about that. I do not see here a summary review of research, but rather a list of issues and possible solutions. Unfortunately, the scope of the issues raised goes far beyond this paper, and therefore I have the impression that the author discusses many things briefly and in a simplistic way. There is probably some explanation for how low-level processes of environmental entropy minimization evolved into high-level cognitive mechanisms, but the outline of this explanation presented by the author seems to me highly incomplete and reductive.

My last point concerns the concept of naturalistic behavior - I failed to understand its meaning. Or was it natural behavior?

I believe that the presented paper in its current form is not ready for publication. Needs significant improvement. I can't quite tell the author in which direction he should go, but it seems to me that it would be good to limit the content a bit and use a good "less is more" heuristic.

References:

Bechtel, W. (2019). Resituating cognitive mechanisms within heterarchical networks controlling physiology and behavior. Theory & Psychology, 29(5), 620–639. https:// doi.org/10.1177/0959354319873725.2020.

Bickhard, M. H. (2003). Process and emergence: Normative function and representation. W: J. Seibt (ed.), Process theories. Cross disciplinary studies in dynamic (121–155). Dordrecht: Springer. https://doi.org/10.1007/978-94-007-1044-3_6.

Bich, L. & Bechtel, W. (2021). Mechanism, autonomy and biological explanation. Biol Philos. 36:53. 1-28. https://doi.org/10.1007/s10539-021-09829-8.

Godfrey-Smith, P. (1996). Complexity and the function of mind in nature. Cambridge: University Press.

Millikan, R. G. (1984). Language, thought, and other biological categories: New foundations for realism. Cambridge: MIT Press.

Moreno, A., & Mossio, M. (2014). Biological autonomy: A philosophical and theoretical inquiry. Dordrecht, the Netherlands: Springer.

Pattee, H. H. (1991). Measurement-control heterarchical networks in living systems. International Journal of General Systems, 18(3), 213–221.

Ruiz-Mirazo, K., & Moreno, A. (2004). Basic autonomy as a fundamental step in the synthesis of life. Artificial Life, 10, 235–259.

Author Response

Dear Reviewer!

I’m very grateful for the careful reviewing and interesting comments, which invoke reexamining of the considered questions.

  • I agree with the note that some argumentations in my paper require links to the previous works in the field, including author’s last publication [99]. In particular, it is true in understanding consciousness in an evolutionary perspective and as the result of natural evolution. Such a claim seems to be trivial, but I believe it to be necessary because one can argue that consciousness is something alike basic quality of matter or, in contrast, it has no any evolutionary meaning at all [5–8]. I do not mention other populist views, which spreading is facilitated by the modern ways of communication. Claiming that consciousness is a result of natural selection assumes that it emerged at a particular point of time in particular circumstances to solve particular tasks of survival. Thus, it is shaped to solve these tasks of species survival rather than represent any ideal rational design. Whereas considering consciousness by analogue with the computer that is still popular today is an attempt to model its functions in the very different circumstances. That is why an interdisciplinary study that comprehends advances in neuroscience and anthropology is needed. Further, claiming human consciousness to be adaptive elaborates on the materialistic understanding of culture, which considers any theory a product of a particular culture that cannot pretend to uncover the ultimate truth. The world-brain alignment (elaborated on in the Temporospatial Theory of Consciousness) is a never-ending process that comes both from the objective reality and actions of the subject.
  • Questioning the difference between the subjective reality and consciousness convinced me to insert another figure from my previous publication. In short, I argue for demarcation between the concepts of subjective reality and consciousness. Subjective reality is a general quality of a system, which functioning and behavior cannot be reduced to the sum of its subsystems activities. Most likely, all organisms with neural system have it, and its complexity varies greatly. In a narrow sense, consciousness is an ability to control the own behavior at the level of intentions. It seems that up to date only humans can master this skill, though some other species probably can be able either. In other words, possibly all living beings have subjective reality, while consciousness is a skill that requires particular complexity of the neural system and must be mastered during the life in the society.
  • Considering interrelation between minimizing entropy and homeostatic mechanisms, I believe the latter represent the tendency of a stable system to last its existence; hence they are directly connected with the survival of individual or species. Maintaining stability in a not so trivial way seems to be an “allostasis”, a very interesting concept, which I decided to avoid because of too much equivocations associated with it. In individual behavior, I can hardly name any activity not associated with the survival. The intelligent use of fire was an evolutionary advantage that was necessary because of the human limited ability to maintain constant body temperature in changing environmental conditions. However, the emerging of culture was constructing the system of higher orderliness, which preservation may require some unnatural behavior from the individual; for example, self-sacrifice in the name of fire. Nevertheless, while existence of this system provides advantages in natural selection, it can last.
  • I am very grateful for mentioning philosophical literature on biological functions, the role of constraints, and biological autonomy. I will incorporate these in my future work. As you justly mention, this particular paper is already overcomplicated to add some more material. I realize that the format of monograph could have been more suitable to consider these issues and synthesize physical systems and society in a less hasty way. However, some points were argued before and I found excessive to repeat them here. Further, in the current conditions articles are able to invoke more vivid discussion than massive books, especially if there is no famous name on the cover of the latter. There is a good discussion with the reviewers, at least.
  • This concerns the question of the assigning this article to which type of document. I believe that the Review must not only summarize published material—because this task is already performed by neuro-networks much better than humans—but to synthesize the most prominent issues in the field and suggest authors theoretical explanation and uncovering new perspectives for the future development. Thus, I agree that this type of work should be qualified as a theoretical research. It can fail, as anyone can. However, the scientific progress and contribution to the field can be defined as falsification of hypotheses.
  • I am not native English speaker, and the concept of “naturalistic behavior” was learned from the other published articles, Pessoa et al. (2022) in particular. Further, there neither were any questions before, nor from the English editing service, nor from the other referees. However, I reexamined this question and found that “naturalistic” can be used in all the cases, though is more suitable when talking about our research of natural behavior (e.g. Kennedy (2022) “The what, how, and why of naturalistic behavior”. Thus, I have changed using of the term to “natural behavior” in some places. Thank you!
  • This article does not have an easy structure and requires keeping in mind too many things. I’m ultimately grateful to you for your work. I believe the value of this research is also in such a style that tries to synthesize a theory that no machine can do yet, just because it expands the constraints of the current thinking in a hasty way. Cutting some part will transform it to something different, and is much more suitable for the new work, during which I will be happy to collaborate.

Sincerely, Ilya A. Kanaev.

Reviewer 2 Report

Thank you for the clarity of the paper, it is very easy to read. I miss a mention in the introduction to some consciousness theories from philosophy of neuroscience such as GWT, IIT or even FEP. How do these theories relate to this paper? I also need you to explicitely state what are your philosophical assumptions regarding consciousness. For example, its origin as an evolutionary process. This is not the case of other consciousness theories such as panpsychism. As the paper is talking about entropy, I think that a formal and analytical (mathematical) definition of entropy is a must. In this sense, I wonder why is the particular parametric expression of entropy used and not other measures of impurity or uncertainty in a random variable like for example the gini criterion. In other words: what is the particular feature of the entropy expression the one that has the fire of consciousness in it? I have strong doubts about that. In this sense, it would be good to clearly state that your vision of consciousness is that it is a phenomenon that can be studied using human understanding (in other words, it resides in our epistemologic space, it is accesible and explainable by us), because some philosophical systems clearly state that phenomenal consciousness is a phenomenon that does not reside under our epistemology, like for example Zen Buddishm philosophy.  Also, you must state which particular entropy are you talking to, there are several definitions, although I assume that is the entropy of theory of information, which is nice to see in this kind of paper! In section 2, you begin with a statement that I found a little bit aggresive to tell you the truth, concretely: "Among the known species, the neural system best represents the integration of body functions and guiding behavior." Mmmm... there are theories that state that plants, that lack a neural system, have some behaviours that are shown to be adaptive to the environment. Why is necessarily the neural system the best? At least, assuming that I agree with you, under which criteria? Another issue is with "examining the neural system using an analog with a computer", there are lots of differences here between both "systems". Maybe a definition of system here is also necessary. A figure about GWT could be helpful in section 2, alternative theories and why GWT is chosen in this paper are necessary to be introduced. "First, the stochasticity of any system comes from the fundamental principles of the
existing reality and thus cannot be eliminated." assuming indeterminism. "Second, entropy, as an intrinsic tendency
to disseminate any orderliness, is vital for an organism's existence because it helps avoid
lodging in one state and returns the homeostatic state from which any required action is
equally possible." Maybe you are talking here not about entropy but about loss of information between the environment and the agent. It is not the same. You must clearly state this in mathematical terms because it is ambiguous and I am afraid that do not understand exactly what are you talking about without a proper analytical definition, including the random variables whose uncertainty about its values is estimated by the entropy criterion.

Another serious issue that I have is that the paper implicitely assumes intelligent behaviour for consciousness (in the sense of awareness or phenomenal consciousness) to flourish. There is no scientific evidence about this nor way to show that this is causally true using the scientific method without assuming that neuroscientific empirical evidence yields causality. There is no way to observe that. You need to specifically include in the paper that you are assuming that the awareness behind phenomenal consciousness emerges as a result of brain activity, which, again, we do not know it for sure.

I think that the paper has future, I have liked it a lot!! But have not read further section as I believe that authors must address my concerns first. So see you in the next iteration!

Author Response

Author’s response:

Thank you so much for you comprehension and sincere attention to the contents of my paper! I appreciate very much your comments and will try to answer point by point.

  1. Introducing theories of consciousness (GWT, IIT, and others) – I’m afraid that if try to summarize these comprehensive theories in few words, it can lead to misunderstanding. Thus, I provided links to a comprehensive monograph [1] and several good reviews of theories of consciousness [7, 33–35]. Further, there may be a set of variable views even within one theory. For example, one can claim that the Global Workspace Theory of consciousness emphasize neural computations, connectivity between brain areas, and activity of the temporal lobe in particular as critical requirements for the arising of consciousness [2, 38– 41]. This understanding can be true according to the most well-known publications by S. Dehaene et. al. However, is it true considering the most recent hypothesis of the brain hominization suggested by J.P. Changeux with coauthors? They promote this prominent theory; nevertheless, one cannot claim that their recent works consider consciousness by analogue with the mechanistic computer [42]. I believe that in many aspects their view became much more complicated and naturalistic. Therefore I emphasized this particular research and its possible connection with the Integrated Information Theory [60–64] that argues for the principle irreducibility of consciousness to a mere sum of the subsystem’s computations. I have tried to demonstrate common currencies of considered theories rather than argue for one of them. This predetermined the current design of the study, where I try to introduce focal points one by one and incorporate them into the framework of the Temporo-spatial Theory. This particular paper is an attempt to elaborate on the materialistic approach and establish closer connections between physical reality and consciousness. The last but not the least, it seems that I should have more extensively refer to my previous publication “Evolutionary origin and the development of consciousness”, Neuroscience and Biobehavioral Reviews 133, 104511 [99], the biggest part of which I devoted to analyzing of the existed theories of consciousness and arguing for understanding consciousness as a particular ability to control behavior at the level of intentions. Possibly, using the final conceptualization of Subjective Reality and Consciousness in Figure 1 is a good idea. Thank you!
  2. Stating my philosophical assumptions regarding consciousness – I’m grateful for this note, it is important since this paper is much more philosophical than empirical study, and my background is in philosophy. In general, I agree with the framework suggested by G. Northoff and his attempt to consider consciousness as a result of the World-Brain alignment. In general, it can be considered as materialism with a focus on the intrinsic activity of the brain. However, it is not Kantian view, because human brain is considered in a context of natural environment – and I believe that my attempts to establish connection between the neural activity and natural evolution can make a contribution to this theory. Further, one needs to decide what materialism is? Is it an attempt to find some simple particles of the matter and reduce all the world to their movement? I don’t think so and agree with V. Hezenberg (“Physics and Philosophy”, 1962) that it is almost have no sense to look for the most simple parts of the matter because this is the question of how much energy we can use and control. He theorized that the matter is different forms of energy, which transformation has objective principles. Hence, materialism is an attempt to find these principles and describe them either in physical events of milliseconds or in evolutionary processes of millions of years. I believe that the concept of entropy as a measure of uncertainty can be the link to bind these different fields of research and constitute a really interdisciplinary framework.
  3. Providing a particular definition of entropy – this also was guided by the materialistic framework, and explanations by K. Marx that any theorizing is grounded on the current level of the culture and restricted by it. If I try to provide some formal or even mathematical definition of the entropy, it will immediately introduce constrains of our current understanding and theorizing. Therefore, I tried to sketch the direction and establish the link between the levels of orderliness in the body and the measurement of the uncertainty in the world-organism relations. As I mention while considering limitations of this paper, finding a formal definition of entropy is a task for the collective of researchers, while any attempts to suggest some definition on the ground of my mere abilities will be voluntary choice of something that I find attractive. I believe that defining entropy as a measure of uncertainty fits this particular study the best. I’ve tried to clarify it, please examine.
  4. Distinguishing phenomenal space (subjective reality) and abilities of the human cognition (consciousness) – I generally agree with the view that almost any organisms may experience their environment – and this is why I’ve tried to distinguish concepts of Subjective reality and Consciousness. Subjective reality emerges in any complex system, which activity overcomes the sum of its subsystem’s activities. I have some doubts if plants fit this criterion, but all animals with neural system definitely do, because interference between signals in the neural system may create a unique pattern that cannot be reduced to activity of its elements. I claim that neural system integrates body functions best because among the known species the ones with the neural system demonstrate better individual abilities for the survival. Neural system provides instant alignment between the world and individual organism, and this alignment is usually limited by the direct effect of the environment and/or learning its regularities (what I described about the anticipation). Further, consciousness is a direct ability to control behavior at the level of intentions rather than just managing consequences of actions that can be simply trained in rats, dogs, and many others. Such ability requires much more complicated neural system, and this is the reason why I highlight the brain hominization hypothesis, studies of the working memory, and social intelligence hypothesis that emphasize the role of social and cultural selection (please see modification of the Figure 1). Thus, I consider consciousness as completely natural but outstanding skill that must be mastered during the lifetime and requires appropriate cultural environment. If not, there is subjective reality, but no ability of the conscious cognition. This allows to consider unique human cognitive abilities in a naturalistic framework while avoiding vulgar reductionism.
  5. If the stochasticity of system implies indeterminism – this question requires much deeper diving in physics and will cause more questions about the mathematical definitions of entropy, which I tried to avoid because of the reasons explained above and my limited abilities. In brief, I assume that determinism is a matter of statistic: the complex thing is quite strictly predetermined by the activity of its subsystems. However, when the system is unified under another physical principle, it obtains much more stability and freedom. This is why neural system usually provides variable behavior, while organisms without it seem to be limited with almost automatic response to the environment. Human brain connectivity is an important issue because it combines activity of the brain areas rather than single neurons. And this is why emerging of culture that tends to consider individuals as subjects of action and responsibility may cause origin of consciousness and self-awareness that provides some freedom to one who can master these skills.
  6. If entropy of the system contributes to its adaptive abilities – it seems that your definition of entropy as a measure of uncertainty is better than mine. This process can be described as a loss of information between the environment and the agent either. Though, I believe this is not the main perspective. I assumed that if some system can be absolutely adaptive to the environment, it implies that the particular conditions are the ultimate causes of its behavior. Hence, when environment changes, its previous adaptive advantages can become the reason for the extinction. This is why any homeostatic system must preserve some degree of uncertainty in its functioning and behavior.

In general, I agree that providing formal or mathematical definition of entropy of measuring uncertainty of the system (or dissemination of the orderliness as I suggested) can contribute this paper a lot. This was my original plan for collaboration – but the pandemic outbreak happened in China during 2022 fall ruined these plans. Because of my abilities’ limitations, I decided to focus on a more philosophical framework and made this theoretical research as a step in the future explaining consciousness in the context of the reality.

As I could understand, the most of comments are caused by the less of clarity in my definition of consciousness. To resolve this uncertainty, I modified the Figure 1 and supplemented it with conceptualization and reference to the previous publication, where I consider subjective reality and consciousness. I believe this contributed to the paper and I am ultimately grateful for your assistance. If you believe that some other and more deep improvements are needed I will be happy to collaborate.

Reviewer 3 Report

This is a thoughtful review article discussing consciousness in the context of entropy and neural evolution. I have a few minor comments to share.

1. In figure 1, it is not fully clear why illusion has a lower entropy of subjective reality as compared to conscious state. Please provide some examples or explanations. In addition, how to measure the entropy of subjective reality, experimentally? 

2. Any existing data that support the opposite relationship between entropy of subjective reality and entropy of neural system? 

3. There is an earlier work by Mashour and Alkire (2013) who discussed consciousness in the context of evolution. The current manuscript may be benefit from relating Top-Down vs. Bottom-Up approaches to consciousness with entropy and orderliness. 

Author Response

Dear Reviewer!
Thank you so much for you sincere attention and appreciation! This is important for me.

  1. My suggestion that illusion has a lower entropy of subjective reality comes from the works of G. Northoff with coauthors, in particular, Northoff, G.; Tumati, S. (2019) “Average is good, extremes are bad”, Neuroscience and Biobehavioral Reviews 104, 11–25. In this research they argue for the inverted relationship between neural mechanisms and functionality of mental features. Further, neural system seems to demonstrate the highest entropy in the state of illusion, while non-conscious states are more predictable from the third-person perspective. The latter is supported by the other studies cited in the text of the paper (it seems that the work of Mashour and Alkire that you suggest in point 3 also mentions the same while considering study by Set et al.). Combining this with the data from the psychiatry that describes many pathological states as creating the subjective world guided by intrinsic principles and other studies allowed to provide such a hypothesis. Further progress can be reached by facilitating studies of consciousness in REM sleep, under the drug effect, and meditative conditions.

Measuring entropy of subjective reality is a very important issue for my work, thank you for reminding to consider it in more details. This is a perspective for the further progress, and one of the main tasks of my research is to set up the foundation for it. On one hand, using report paradigm to describe states of the subjective reality can be the easiest way. On the other, I would like to avoid such methods since they seem to be a voluntary interpretation against which I argued before. Hence, I assume that using non-report paradigm is preferable. However, how to obtain the access to the subjective states? I believe that it can be reached by combing advances in studying correlation between one’s ability for the self-control, modifying neural activity by extracranial stimulation, drugs, and conscious dreaming – with objective neuroimaging described as brain-to-brain synchrony by Yeshurun et al. (2021), and Koban et al. (2021). In other words, combining objective neuroimaging with performing naturalistic activity must contribute to the framework of the world-brain alignment suggested in the Temporo-Spatial Theory of Consciousness. This is the new horizon for the behavioral neuroscience and materialistic understanding of consciousness that can provide theoretical foundation for ideals of the moral responsibility, creativity, and free will.

  1. Considering entropy as a conceptual link between the neural system’s activity and subjective reality is an attempt to elaborate on the previous theories and advances in science. Thus, it is a reinterpretation that must prove its theoretical power in the future experiments. I define entropy in as a tendency to dissolve orderliness of the system with time; it can be considered as a measure of the uncertainty either. Epistemologically, entropy measures our ability to predict future states of the system. Below, I provide several issues that allowed me to make such a conclusion:
    1. It was demonstrated that within the non-conscious states of the subjective reality one’s neural system has lower entropy, while increasing the level of awareness and self-control makes neural activity more unpredictable from the third-person point of view. However, conscious state is qualified as much more well-organized and systematic than non-conscious state. Thus, one can assume that entropy of the conscious state is lower than in unconscious state (anyone can compare awareness with dreaming). However, it is critical to consider conscious state in light of alignment with the world, which provides regularities and guides entropy of the neural system (as I tried to discuss in the paper). In a state of illusion one’s subjective reality much more depends on their neural system’s activity and learned patterns, up to losing any connection with the objective reality. Thus, unconstrained increasing own orderliness of the system may lead to the ultimate disorderliness and death. There must be a balance between the world and intrinsic activity of the neural system to maintain stability.
    2. Studies in working memory provide a profound material to bind neural activity with individual behavior. In particular, I find very promising the flexible memory hypothesis. It suggests that working memory limitations arise from the interference of the neural sub -systems activities. Thus, preserving orderliness of the consciousness becomes harder at each step of complication, until neural system just reaches its limits. The direct connection between working memory load and self-control demonstrates this best. Using transcranial stimulations and other methods to temporarily restore working memory abilities provide and external support to the neural system and most likely decrease its entropy caused by aging or other circumstances.
    3. The social intelligence hypothesis suggests that hominization of the brain and obtaining ability for the conscious actions could be caused by social selection and emerging of culture. Dunbar (2020) suggested that many cultural practices (mystical stance, dance, song, art) emerged as an attempt to decrease the stress from the co-living in a big group and additional methods to support social cohesion that is critical for human survival. One can consider these practices somewhat similar to extracranial stimulation discussed above since they serve as an external support for individual activity and behavior. Further development of society and culture caused emerging of consciousness as an ability to control behavior in light of intentions. This self-control task is very resource-consuming and requires constant support. However, it provides the highest orderliness of the subjective reality among the known ones. It seems that the capacities of individual neural system are not enough, and this is why the effect of the society is needed.
  2. Thank you for the suggestion to consider this work by Mashour and Alkire! This response to my article found me in a business trip in a rural area, and I cannot check my library. I am sure that I have read works by Mashour (though, maybe not exactly this one), and aware about Bottom-Up and Top-Down distinctions. Possibly, it was a mistake not to consider them more explicitly. The reason was to avoid extensive “labeling” that can foreshadow original meanings in some cases. I have tried to present a comprehensive synthesis of theories, while providing demarcation between these two frameworks will require deeper analysis of data in neuroscience that would complicate this text and could be beyond my current abilities. Basing on your comments, I decided to modify Figure 1 and supplemented with my conceptualization of subjective reality and consciousness from, the previous publication. I hope that this can make some statements more clear – including the distinction between Bottom-Up and Top-Down paradigms. I agree with the considered authors that the experiencing emerged at a very basic form of organism’s evolution; while consciousness is a skill that requires very complicated neural structure and incorporation in the society. This question deserves much deeper elaboration than I can perform during the paper revision, and thus requires the new study, during which I will be happy to collaborate!

Sincerely, Ilya A. Kanaev

Round 2

Reviewer 1 Report

Dear Author,

Thank you very much for explanations. I think some issues still require further study, but I understand that you will continue your research. The issues raised are very ambitious and therefore susceptible to criticism or various objections. However, I think you are basically right. I accept the paper as it stands and wish you good luck in your further research.

Reviewer 2 Report

Thanks for the reply and changes! My main observations have been covered. Although I believe a mathematical representation would enrich even more this paper, it can be something done for further work (if interested, please contact).

Reviewer 3 Report

I thank the author to address my concerns.